# ESE-YOLOv8: A Novel Object Detection Algorithm for Safety Belt Detection during Working at Heights

**DOI:** 10.3390/e26070591

**Published:** 2024-07-11

**Authors:** Qirui Zhou, Dandan Liu, Kang An

**Affiliations:** 1The College of Information, Mechanical and Electrical Engineering, Shanghai Normal University, Shanghai 201412, China; zhouqirui@mail.shiep.edu.cn; 2College of Electronics and Information Engineering, Shanghai University of Electric Power, Shanghai 201306, China

**Keywords:** YOLOv8, object detection, attention mechanism, safety belt detection, information entropy

## Abstract

To address the challenges associated with supervising workers who wear safety belts while working at heights, this study proposes a solution involving the utilization of an object detection model to replace manual supervision. A novel object detection model, named ESE-YOLOv8, is introduced. The integration of the Efficient Multi-Scale Attention (EMA) mechanism within this model enhances information entropy through cross-channel interaction and encodes spatial information into the channels, thereby enabling the model to obtain rich and significant information during feature extraction. By employing GSConv to reconstruct the neck into a slim-neck configuration, the computational load of the neck is reduced without the loss of information entropy, allowing the attention mechanism to function more effectively, thereby improving accuracy. During the model training phase, a regression loss function named the Efficient Intersection over Union (EIoU) is employed to further refine the model’s object localization capabilities. Experimental results demonstrate that the ESE-YOLOv8 model achieves an average precision of 92.7% at an IoU threshold of 50% and an average precision of 75.7% within the IoU threshold range of 50% to 95%. These results surpass the performance of the baseline model, the widely utilized YOLOv5 and demonstrate competitiveness among state-of-the-art models. Ablation experiments further confirm the effectiveness of the model’s enhancements.

## 1. Introduction

At various engineering sites, including construction sites and power installations, workers are tasked with performing duties at elevated locations, commonly referred to as ‘working at height’. These operational scenarios inherently expose workers to risks, underscoring the importance of wearing safety belts as required by regulations. However, due to various factors, some workers might choose not to wear or may even remove their safety belts while working at heights. Such actions can lead to fall accidents, posing a threat to the well-being of both workers and ground personnel.

Therefore, supervisors are required to be on-site or via surveillance cameras to monitor and promptly remind workers not complying with safety belt requirements through intercoms or broadcast systems. However, conventional monitoring methods require continuous attention from supervisors, who must observe all workers engaged in tasks at height simultaneously, posing a challenge to their attentiveness and focus.

To address this challenge, using cameras and deep learning-based object detection algorithms for automated safety belt detection presents a feasible solution. In recent years, the development of deep learning has produced numerous object detection algorithms. However, in the context of safety belt detection for workers engaged in high-altitude operations, new methods are scarce. Existing solutions still rely on outdated algorithms. Previous algorithms focused more on adapting to low-performance terminal devices or portable equipment at construction sites, prioritizing real-time algorithm performance at the cost of accuracy. As construction sites become increasingly intelligent and digitized, the landscape is shifting toward adopting multi-camera configurations and centralized processing on high-performance servers. This suggests the potential for employing larger models or their variants.

In light of the current landscape, this paper introduces an enhanced deep learning network, ESE-YOLOv8, specifically designed to determine whether workers engaged in high-altitude tasks are wearing safety belts. The main contributions of this study are as follows:The Integration of Efficient Multi-Scale Attention (EMA): EMA is incorporated into the backbone network of YOLOv8 to optimize feature extraction. EMA achieves cross-channel interaction and increases information entropy by employing cross-spatial learning through two branches with different convolution kernels. It encodes spatial information into channels via pooling, avoiding potential information loss by not reducing channel dimensions.Slim-Neck Configuration with GSConv: In the neck of the network, GSConv and its associated modules replace the original convolutional operations and the C2f module. This slim-neck configuration integrates depth-wise separable convolutions with standard convolutions, thereby reducing computational load without diminishing the information entropy generated by the convolutions.Efficient Intersection over Union (EIoU) Loss Function: To improve detection accuracy, the loss function is replaced with EIoU. EIoU directly compares the width and height differences between the predicted bounding box and the ground truth, resulting in more precise localization and faster convergence.

The enhanced ESE-YOLOv8 network is evaluated against its original version, the widely used YOLOv5, and four other state-of-the-art models. Extensive ablation experiments corroborate the performance improvements achieved through these enhancements.

This comprehensive approach ensures that the ESE-YOLOv8 model not only maintains high accuracy and efficiency but also significantly advances the current capabilities of real-time object detection in safety-critical applications.

## 2. Related Work

### 2.1. Deep Learning

Deep learning-based object detection algorithms are generally categorized into two main types: one-stage and two-stage approaches. One-stage algorithms include the You Only Look Once (YOLO) series [1,2,3,4] and SSD [5], which are regression-based detection algorithms. They directly use pre-trained backbone networks to generate class probabilities and positional coordinates of detected objects. While their detection precision slightly lags behind that of two-stage object detection algorithms, they exhibit faster detection speeds. Two-stage algorithms include R-CNN [6], Fast R-CNN [7], Faster R-CNN [8], and Mask R-CNN [9], which are region proposal-based detection algorithms. These algorithms first generate candidate regions through RPN networks for target detection, followed by classification and position regression for these regions. Because these algorithms divide object detection into two steps, their detection speeds are lower compared to those of one-stage algorithms.

Because of their reduced complexity, one-stage algorithms are faster than two-stage algorithms and can be deployed on various low-cost, low-performance terminal devices. Among them, the YOLO series of algorithms, introduced in 2015, has spawned numerous versions and variants, gaining significant attention from both researchers and engineers. The YOLOv5 model, released by the Ultralytics team on 1 April 2020, offers multiple variants that balance accuracy and speed, providing users with a range of choices. Two years later, the same team released a new model, YOLOv8.

### 2.2. Security Applications of Object Detection

Object detection algorithms are widely used for security-related applications, with a significant emphasis on detecting safety helmets on construction sites and identifying the unsafe behaviors of various vehicle drivers. Xu and colleagues [10] introduced a module for aggregating semantic context information and a feature selection fusion structure in CenterNet, achieving an 87.21% mAP in safety helmet detection, surpassing the baseline by 5.12%. Li et al. [11] combined an improved YOLOv5 with StrongSORT for detecting and tracking whether workers are wearing safety helmets, achieving higher accuracy than the baseline and three other models. Ni and Hu [12] introduced K-means clustering into YOLOv5 and adjusted the network’s loss function, improving the accuracy in detecting safety helmets and reducing the false negative rate for head detection to 3.16%.

Tai [13] and others enhanced YOLOv5’s generalization and accuracy in detecting safety helmets under different weather conditions through data augmentation, the K-DAFS dynamic anchor box mechanism, and attention mechanisms. Jayanthan and Domnic [14] used a CNN-based encoder to extract features and a Transformer-based decoder to identify whether motorcycle drivers were wearing helmets, achieving higher accuracy than four other models. They also demonstrated good adaptability in general object detection tasks beyond helmet detection. Almazroi et al. [15] employed a CNN for detecting eyes and mouths as facial organs to determine if drivers were distracted or drowsy, and to detect if drivers were wearing seatbelts. They achieved high accuracy in both detection tasks, issuing alerts in cases of poor driver mental states or failures to wear seatbelts.

Hosseini and Fathi [16] used YOLOv5 to detect the position of car windows and an improved ResNet34 to classify whether drivers are wearing seatbelts, achieving the highest accuracy among several concurrent methods. In the field of detecting unsafe behaviors during work at heights, scholars have also opted for deep learning approaches. Liu [17] employed AlphaPose to extract skeletal key point coordinates and a spatiotemporal graph convolutional network (ST-GCN) for action recognition. Chen [18] used YOLOv5 for worker posture estimation and introduced an automated annotation module for labeling motion status data, thereby reducing manual annotation costs. Both methods identified unsafe ladder climbing actions among on-site workers, but were unable to detect unsafe actions when operating on elevated platforms.

Hu [19] used Faster R-CNN to detect unsafe actions of workers during high-altitude operations, although it did not account for situations where certain actions deemed to be unsafe might actually be safe and conducive to work, especially when workers are wearing safety belts. Both Zhang [20] and Dun [21] adopted YOLOv4 for safety belt detection, with the smaller variants offering rapid processing speeds. However, these approaches compromised on accuracy, potentially leading to higher error rates due to frequent misclassification.

## 3. Methods

This section introduces the overall structure of the ESE-YOLOv8 model. Subsequently, the baseline YOLOv8 model, the EMA mechanism for accuracy improvement, the modified neck, and the enhanced loss function will be individually detailed. The architecture of the proposed model is illustrated in Figure 1.

The model processes images in the form of a 3 × 640 × 640 array. For images of other sizes, the model preprocesses them by either compressing or stretching to match the standard dimensions. Before the Spatial Pyramid Pooling Fast (SPPF) module, the image undergoes feature extraction through the Conv and C2f modules. The EMA module biases the extraction of channel information, emphasizing crucial data. The SPPF module mitigates image distortion issues from cropping and scaling, preventing the redundantly correlated feature extraction that is common in convolutional neural networks. This enhances the model’s speed. Following the SPPF module, the image is segmented into three scales: 128 × 80 × 80, 256 × 40 × 40, and 512 × 20 × 20, through various up-sample modules and GSConv and VoV-GSCSP modules for down-sampling.

Throughout the model, all Conv modules in the detection head and the C2f and SPPF modules use 2D convolutions. The Conv modules in the C2f and SPPF modules, along with those in the detection head, utilize a 1 × 1 convolutional kernel. Other Conv modules, including those in the Bottleneck modules and the detection head, employ 3 × 3 convolutional kernels. Details on the newly introduced EMA, GSConv, VoV-GSCSP, and the modified loss functions will be detailed in subsequent sections.

### 3.1. YOLOv8

Our model is based on YOLOv8, an object detection algorithm developed by the Ultralytics team [22]. The key features of the model are outlined below:YOLOv8 employs the C2f module. As depicted in Figure 1, the C2f module is composed of 2 convolutional modules and a stack of n Bottleneck modules where each Bottleneck module comprises 2 convolutional modules. In the YOLOv8 used in this study, specifically for C2f2 and C2f3, n is assigned the value of 2; for the remaining C2f modules, n is set to 1. The C2f module includes both the concatenated main branch of Bottlenecks and parallel connections via concatenation. This configuration creates multiple branches of gradient flow, enhancing precision through the acquisition of diverse gradient flow information.It employs a distinct decoupled head structure that separates classification and detection heads, enabling the simultaneous learning of distinct tasks such as bounding box regression and category prediction. The learning processes for these tasks are not affected by mutual interference, allowing for independent optimization during refinement.It adopts an Anchor-Free mode that is not affected by scale variations and effectively detects small objects, unaffected by the lack of specially designed anchor boxes. This is ideal for our targeted scenarios where there can be significant scale variations between people and safety belts appearing simultaneously. In some cases, distant workers may appear as small-scale objects within the entire image.

As an outstanding model in recent years, YOLOv8 still has room for improvement in terms of accuracy. Therefore, we have introduced two enhancements that will be discussed in the following sections.

### 3.2. Efficient Multi-Scale Attention (EMA)

In object detection tasks, spatial and channel attention mechanisms play a crucial role in feature extraction from images, as they enable inter-channel interactions, establishing connections between different channels and expanding the amount of information obtained from the input. This effectively increases the model’s information entropy, allowing it to learn more complex or subtle patterns. However, some attention mechanisms reduce the dimensionality of channels during inter-channel interactions, which can lead to information loss and a decrease in information entropy, making it more difficult for the model to learn. The research by Wang et al. [23] confirmed the benefits of inter-channel interactions and the drawbacks of dimensionality reduction. Our model incorporates the EMA mechanism. Proposed by Daliang et al. [24], this mechanism mitigates the adverse effects of dimensionality reduction across channels on a deep visual representation during the modeling of inter-channel relationships. Figure 2 illustrates the structure of the EMA module.

The module initially divides any given input feature map $y\in R^{H\times W\times C}$ into several sub-feature maps along the channel direction (C). Once grouped, the feature map undergoes the extraction of attention weight descriptors via three parallel routes. Two routes employ a 1 × 1 convolutional kernel and the third utilizes a 3 × 3 kernel. The 1 × 1 branch encodes the channels’ width (W) and height (H) through two 1D global average pooling operations, effectively capturing positional information along their vertical dimensions [25]. These results are concatenated along the H and processed through a 1 × 1 convolution. It is important to note that in this 1 × 1 convolution, no dimensionality reduction is performed to avoid potential information loss, as seen in other attention mechanisms such as Coordinate Attention (CA) [25], which can degrade model performance. The convolution output is split into two vectors, which are then shaped by non-linear sigmoid functions to fit a two-dimensional binomial distribution. These vectors are subsequently multiplied, facilitating cross-channel feature interaction. The 3 × 3 branch utilizes a 3 × 3 kernel to capture multi-scale feature representations. Following cross-spatial learning, vectors from both branches are transformed into 1 × C//n and WH × C//n configurations, then processed through 2D global average pooling to encode global spatial information. The average pooling output undergoes a linear transformation using a softmax function, which is then multiplied with the vector from the opposite branch to produce two spatial attention maps for this group.

Finally, the output features of each group are mapped based on the set of generated spatial attention weights, using a sigmoid function. These weights illustrate the attention mechanism’s emphasis on different feature information, derived from the process of reducing the information entropy of the fused feature information and enhancing information transmission efficiency. The input and output shapes of the entire EMA module are identical, providing a pixel-level global context for the feature map. In this paper, the EMA modules are inserted before layers 2, 4, 6, and 8, between the Conv module and the C2f module, with the group count (n) set to 8.

### 3.3. Slim Neck

As the feature map progresses through the neural network’s backbone to the neck, it reaches its maximum channel capacity while its width and height are minimized. Hulin et al. [26] developed GSConv, a lightweight convolution for the neck that maintains hidden channel connections without increasing the FLOP count. Figure 3 depicts the structure of GSConv.

In this structure, DW refers to depth-wise separable convolution, a type of channel-sparse convolution that requires less computation than standard convolutions, but at the cost of lower performance. This is because DW Conv performs convolutions on spatial and channel information separately, without generating cross-channel information, resulting in lower information entropy compared to standard convolutions. By stacking DW Conv with standard convolutions and performing a shuffle operation on their outputs, the model can leverage the low computational cost of DW Conv and the performance of standard convolutions. GSConv employs linear operations for shuffling to adapt to various runtime environments, while its computational load is only 50% of that of standard convolutions. In this model, the kernel size for Conv is k, and for DW Conv, it is 5. Figure 4 illustrates the structure of VoV-GSCSP, constructed from GSConv. The kernel size for Conv is 1, and for the two GSConvs, the k is 1 and 3, respectively, from top to bottom.

The neck of YOLOv8 is reconstructed by replacing the C2f and Conv blocks with VoV-GSCSP and GSConv, respectively, as illustrated in Figure 1. According to the literature [26], this redesigned neck reduces the model’s computational load and enhances accuracy.

### 3.4. Loss Function

The YOLOv8 loss function includes two components: category loss and bounding box regression loss. Category loss pertains to the model’s classification accuracy, and bounding box regression loss reflects its accuracy in object localization. Cross-entropy loss is applied to category loss, while bounding box regression loss utilizes Distribution Focal Loss (DFL) and Complete Intersection over Union (CIoU) loss. Given our focus on scenarios with a limited number of categories, we emphasize refining bounding box regression loss to enhance the model’s localization capabilities.

The Intersection over Union (IoU) is a measure of the area intersection divided by the area union between the predicted bounding box and the ground truth bounding box [27]. Its mathematical expression is given by Equation (1):(1)$$\mathrm{IoU}=\frac{A\cap B}{A\cup B}$$
where A represents the predicted bounding box, and B represents the ground truth bounding box.

The expression for the IoU loss function is given by Equation (2), which quantifies the similarity between A and B:(2)$$L_{\mathrm{IoU}}=1-\frac{\left|A\cap B\right|}{\left|A\cup B\right|}$$

The limitation of the IoU loss is that when there is no overlap between bounding boxes A and B, the IoU remains constant at 0, failing to reflect the proximity or distance between the two boxes. Additionally, the IoU loss function exhibits a slower convergence speed.

To address these issues, Rezatofighi et al. introduced the Generalized IoU (GIoU) loss function [28], which is defined by Equation (3):(3)$$L_{\mathrm{GIoU}}=1-\mathrm{IoU}+\frac{\left|C-(A\cup B)\right|}{\left|C\right|}$$
where C is the smallest bounding box that contains both bounding boxes A and B. It captures the distance between A and B even when they do not overlap. However, when one box fully encloses another, this loss function degenerates into the IoU loss function, still exhibiting the issue of slow convergence.

The CIoU loss function [29] used in YOLOv8 is expressed as Equation (4):(4)$$L_{\mathrm{CIoU}}=1-\mathrm{IoU}+\frac{\rho^{2}(b,b^{\mathrm{gt}})}{c^{2}}+\alpha \nu$$
where b represents the center point of the predicted bounding box, b^gt^ represents the center point of the ground truth bounding box, $\rho (b,b^{\mathrm{gt}})$ represents the Euclidean distance between these two points, and c represents the diagonal length of the smallest enclosing box that contains both predicted and ground truth boxes. ν is given by Equation (5), and α is given by Equation (6):(5)$$\nu =\frac{4}{\pi^{2}}{\left(\arctan\frac{w^{gt}}{h^{gt}}-\arctan\frac{w}{h}\right)}^{2}$$
(6)$$\alpha =\frac{\nu }{(1-\mathrm{IoU})+\nu }$$

The CIoU considers three key geometric factors: the overlap area, the center point distance, and the aspect ratio. Despite its high accuracy, it overlooks cases where the predicted box’s width and height change proportionally, leading to an increased disparity with the ground truth box. When there is a difference in the width and height between the true bounding box and the predicted bounding box, but their aspect ratios are very close, the CIoU becomes less sensitive. This can result in slow convergence and insufficient localization accuracy. The EIoU loss function [30], proposed by Zhang et al., is expressed as Equation (7):(7)$$\begin{array}{l} L_{\mathrm{EIoU}}=L_{\mathrm{IoU}}+L_{dis}+L_{asp}\\ =1-\mathrm{IoU}+\frac{\rho^{2}\left(b,b^{\mathrm{gt}}\right)}{{\left(w^{c}\right)}^{2}+{\left(h^{c}\right)}^{2}}+\frac{\rho^{2}\left(w,w^{gt}\right)}{{\left(w^{c}\right)}^{2}}+\frac{\rho^{2}\left(h,h^{gt}\right)}{{\left(h^{c}\right)}^{2}} \end{array}$$
where $w^{c}$ and $h^{c}$ are the width and height of the smallest bounding box C that contains both bounding boxes A and B. The loss function is divided into three parts: the IoU loss $L_{IoU}$, distance loss $L_{dis}$, and aspect loss $L_{asp}$. The advanced features of the CIoU are preserved in this formulation. Furthermore, the EIoU directly reduces the differences in width and height between the target box and anchor box, resulting in accelerated convergence and improved localization accuracy.

## 4. Experiments

### 4.1. Dataset

The dataset used in this study originates from a power plant located in Guangdong, China. Sample images and their corresponding labels are illustrated as shown in Figure 5.

The dataset consists of a total of 2016 images, annotated for ground personnel, aerial personnel, and safety belts. For this study, 1269 images are designated as the training set, 142 images as the validation set, and the remaining 605 images as the test set.

### 4.2. Evaluation Metrics

In this study, the overall Mean Average Precision (mAP) is used to evaluate the model’s performance. The mAP is determined by averaging the average precision (AP) across each target class. The AP is calculated based on precision (P) and recall (R), and the formulas for these metrics are in Equations (8)–(11):(8)$$P=\frac{T_{P}}{T_{P}+F_{P}}$$
(9)$$R=\frac{T_{P}}{T_{P}+F_{N}}$$
(10)$$\mathrm{AP}=\sum_{i=1}^{n-1}(R_{i+1}-R_{i})\cdot \frac{P_{i+1}+P_{i}}{2}$$
(11)$$\mathrm{mAP}=\frac{1}{N}\sum_{i=1}^{N}{\mathrm{AP}}_{i}$$
where $T_{P}$ represents the number of true positive samples correctly detected, $F_{P}$ represents the number of false positive samples incorrectly classified as positive from negative samples, and $F_{N}$ represents the sum of false negatives and false positives that were incorrectly classified as negative. AP is the area under the precision-recall (P-R) curve, where R is plotted on the x-axis and P on the y-axis for different confidence thresholds. mAP50 represents the overall Mean Average Precision at an IoU threshold of 50%. mAP50-95 represents the overall Mean Average Precision across different IoU thresholds ranging from 50% to 95%.

### 4.3. Environment and Model Parameters

The experiments were performed on a 64-bit Windows 11 operating system with an NVIDIA GeForce RTX 4090 D GPU and an AMD Ryzen 7 7800X3D CPU. These experiments were executed within the PyTorch 1.13 framework. The model was trained for a total of 200 epochs, with a batch size of 64. The initial learning rate was set to 0.01 and was gradually reduced to 0.00001. The input image size was set to 640 × 640 pixels.

### 4.4. Results

A comparison of the model proposed in this paper, YOLOv8, the commonly employed YOLOv5, and four SOTA models is depicted in Table 1. For comparing SOTA models, YOLOv6 3.0 [31], Gold-YOLO [32], YOLOX and ATSS [33], we choose variants that have parameter and Floating-Point Operation (FLOP) complexities closer to the variant of our model variant, small(s), in this paper.

Table 1 indicates that the model proposed in this paper achieves a high mAP. The mAP50 of ESE-YOLOv8 is 92.7%, which is 1.5%, 1.3%, 3.6%, 0.9%, and 16% higher than YOLOv5, YOLOv8, YOLOv6 3.0, Gold-YOLO, and ATSS, respectively. Meanwhile, the mAP50-95 of ESE-YOLOv8 is 75.7%, which is 10.7%, 0.9%, 3.4%, 2.4%, and 32.7% higher than YOLOv5, YOLOv8, YOLOv6 3.0, Gold-YOLO, and ATSS, respectively. Compared to the model proposed, YOLOX demonstrates a 0.2% higher mAP50, an 8% lower mAP50-95, a reduction of 1.3 M parameters, and an increase of 0.7 G FLOPs. This suggests that, at lower IoU thresholds (i.e., coarser localization), YOLOX has a greater advantage, while at higher IoU thresholds (i.e., more precise localization), our model excels. Considering that YOLOX employs an additional data augmentation technique called MixUp and decouples its detection head into three tasks—classification, foreground-background discrimination, and bounding box prediction, with an extra branch for foreground-background discrimination compared to our detection head–these two aspects might contribute to its higher mAP50 and could be considered as potential areas for our future research.

Examples of the proposed model’s detection results are shown in Figure 6. As observed from Figure 6, our model exhibits high accuracy and effective detection results.

## 5. Ablation Study

In order to explore the impact of each individual improvement proposed in this paper on the performance of the YOLOv8 model, this section conducts an ablation study on the ESE-YOLOv8 model. Table 2 presents the performance changes of the ESE-YOLOv8 model when each improvement is removed individually.

In the table, √ indicates that the network includes the corresponding structure in the experiment, and × indicates that it does not. From Table 2, it can be observed that when the slim neck is removed, the mAP50 decreases by 0.3%, while the parameter counts and FLOPs increase by 0.9 M and 2.7 G, respectively. Upon removing the EIoU as well, the mAP50 further decreases by 0.3%, and the mAP50-95 decreases by 0.2%. Subsequently, when the EMA is removed, the mAP50, mAP50-95, parameter count, and FLOPs decline by 0.7%, 0.7%, 0.1 M, and 1 G, respectively.

The heatmaps before and after integrating the EMA into the baseline model are shown in Figure 7. We used Grad-CAM, proposed by Ramprasaath et al. [34], to create the heatmaps. For the original backbone without EMA, we opted to generate a heatmap for layer 7, while for the backbone with EMA, we chose layer 11, where the last EMA is located.

In Figure 7, the bright blocks within the dark background represent the focus points of the model during detection, with a shift towards red indicating a higher level of focus. From Figure 7, it can be observed that before incorporating EMA, the model’s attention points are relatively dispersed, while after incorporating EMA, the attention points are more concentrated and closer to the human body and safety belts. This suggests that the inclusion of EMA, after integrating inter-channel and spatial information, enhances information entropy, optimizing learning outcomes from richer information, and allows the model’s backbone to allocate weights to feature information more rationally, improving the transmission efficiency of key feature information and reducing the interference of irrelevant information.

After incorporating EMA, the heatmaps before and after adding the slim neck are shown in Figure 8. To understand the impact of the neck on the model, we selected layers 19, 22, and 25 for analysis and created heatmaps, as these layers are directly connected to the detection head within the neck.

Comparing Figure 8a with Figure 7, it is evident that the model’s attention points, aggregated by the EMA block, become more dispersed after passing through the neck but remain focused near the target, avoiding the situation seen in Figure 7a. After incorporating the slim neck, the model’s attention becomes more focused on the targets that need to be detected, as illustrated in Figure 8b.

When the CIoU and EIoU are employed, the detection performance of YOLOv8 is depicted in Figure 9.

From Figure 9, it is evident that the utilization of the EIoU results in higher detection accuracy when compared to the CIoU. Employing the EIoU not only improves detection accuracy but also enables the detection of objects that might have been overlooked, such as the safety belt in Figure 9a.

## 6. Conclusions

This paper presents ESE-YOLOv8, an innovative object detection algorithm optimized for detecting the use of safety belts by workers. Leveraging the YOLOv8 architecture, EMA modules are strategically integrated before the 2nd, 4th, 6th, and 8th layers to enhance the extraction of feature information. The model’s neck has been redesigned as a slim neck utilizing GSConv, and the conventional CIoU loss function has been replaced with EIoU to alleviate issues hindering loss reduction and to expedite convergence. Experimental evaluations indicate that the proposed method outperforms the baseline, the widely utilized YOLOv5 in engineering applications, and exhibits competitive performance compared to state-of-the-art approaches. Ablation experiments are conducted to systematically analyze the impact of each refinement on model accuracy.

However, our research also identifies areas necessitating further exploration. Notably, the use of box annotations for safety belts results in a low pixel count within the bounding box, which means that the effective information content in the annotations is very low. This can lead to inefficient learning due to low information entropy, or cause overfitting to some irrelevant background areas. We are considering alternative annotation strategies to increase the information content of the annotations, improve the dataset quality, or adopt methods to filter out noise to obtain useful information. Moreover, the superior performance relative to YOLOX highlights the potential impact of varied data augmentations and background–foreground distinctions. Future efforts will focus on extensive research and the refinement of data augmentation methods and detection head architectures to enhance detection accuracy and robustness.

## Figures and Tables

**Figure 1 entropy-26-00591-f001:**
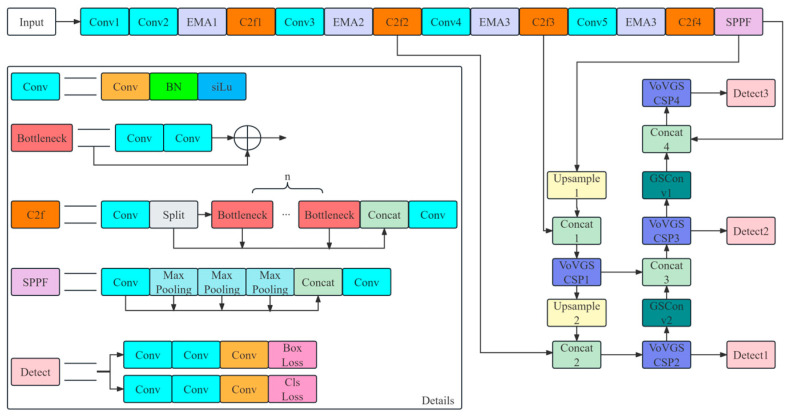
ESE-YOLOv8 model architecture.

**Figure 2 entropy-26-00591-f002:**
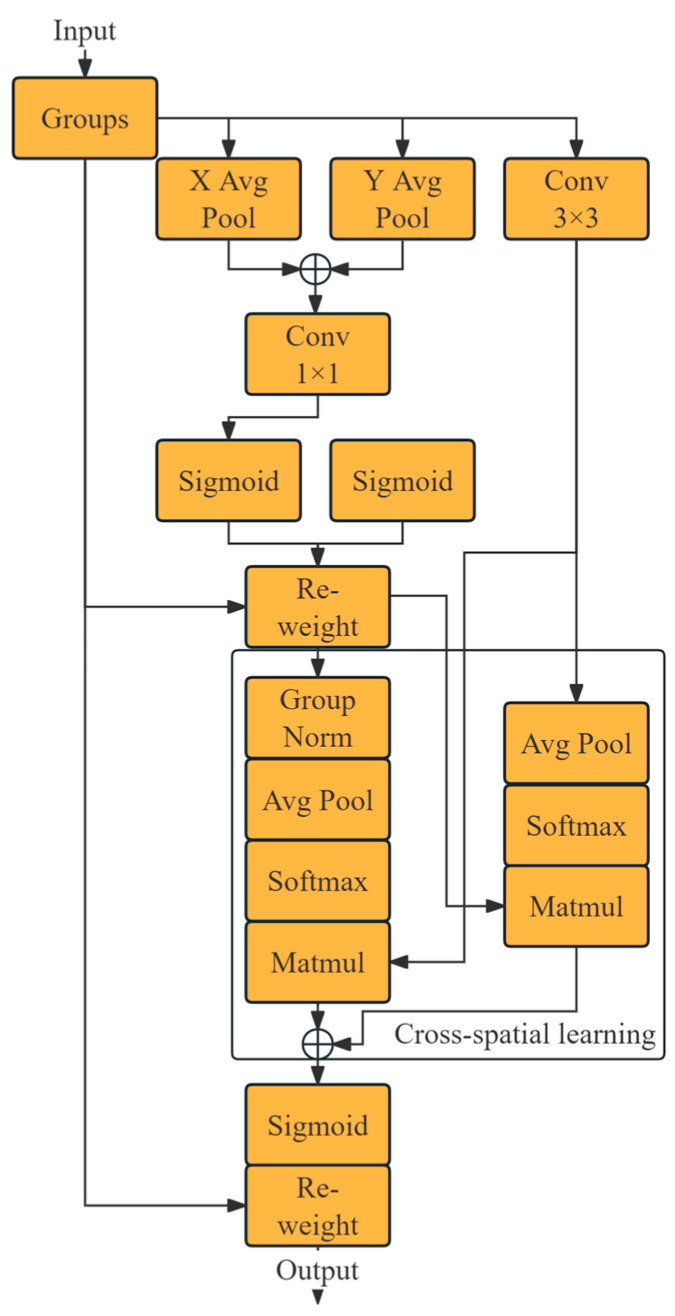
Efficient Multi-Scale Attention (EMA)’s block architecture.

**Figure 3 entropy-26-00591-f003:**
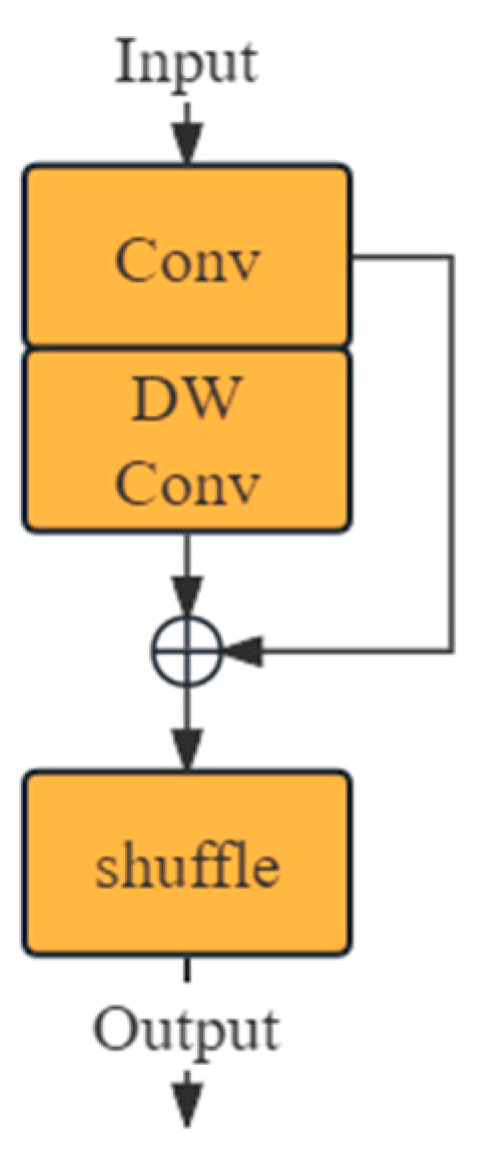
GSConv’s architecture.

**Figure 4 entropy-26-00591-f004:**
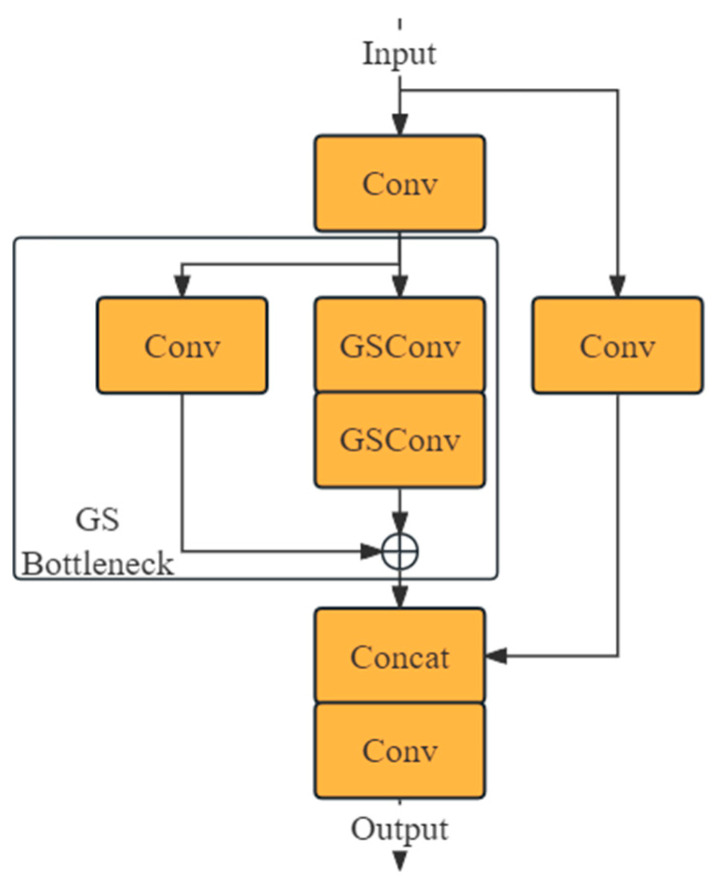
VoV-GSCSP’s block architecture.

**Figure 5 entropy-26-00591-f005:**
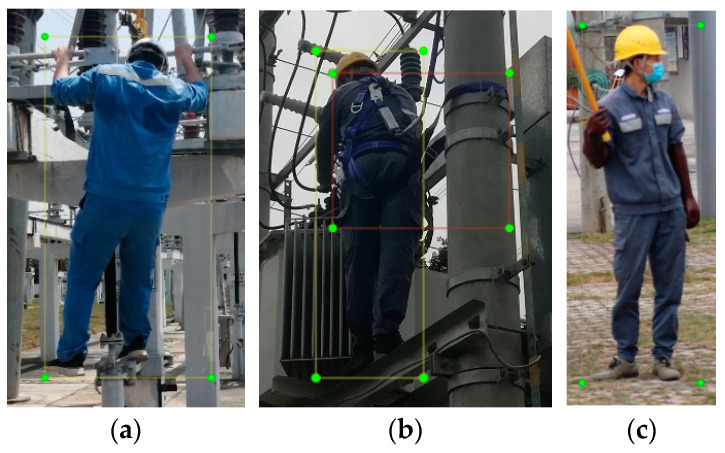
Examples of dataset images and labels. (**a**) An aerial person label; (**b**) an aerial person with a safety belt label; (**c**) a ground person label.

**Figure 6 entropy-26-00591-f006:**
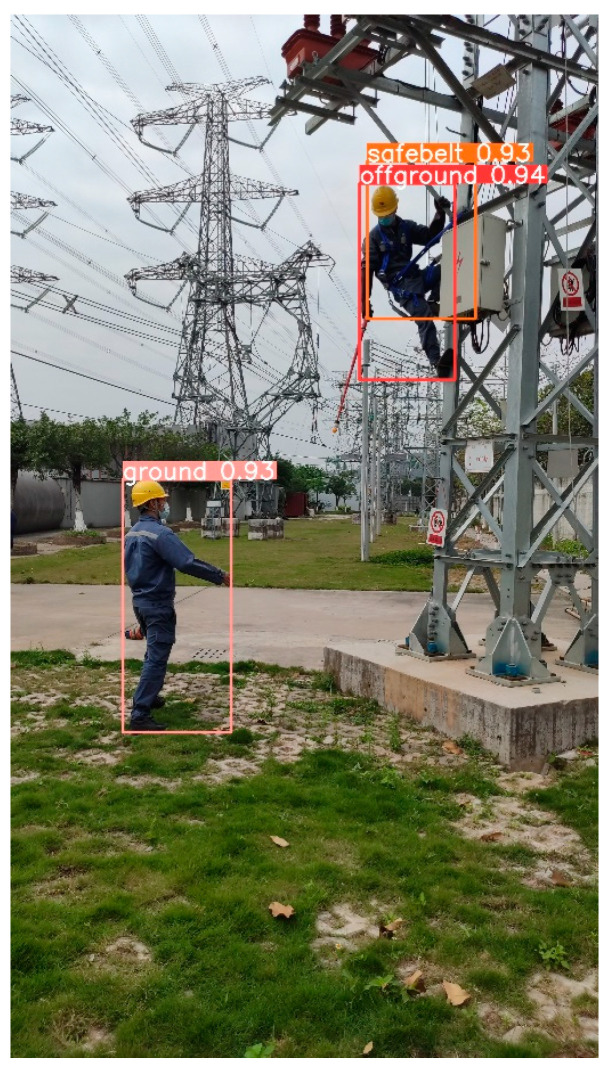
A detection result.

**Figure 7 entropy-26-00591-f007:**
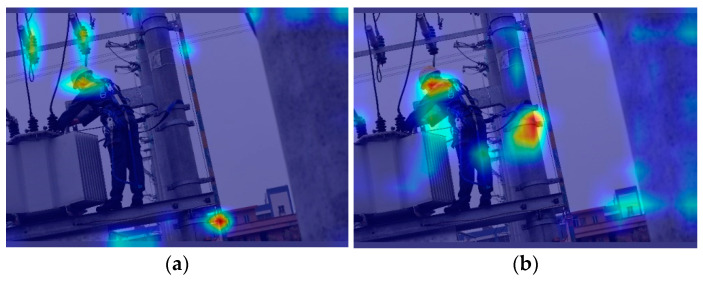
Impact of the EMA block on a heatmap. (**a**) Without the EMA block; (**b**) with the EMA block.

**Figure 8 entropy-26-00591-f008:**
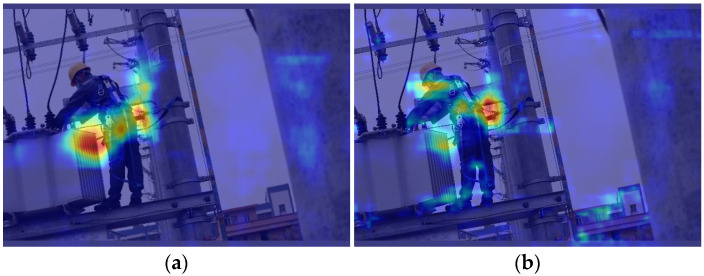
Impact of a slim neck on a heatmap. (**a**) Without a slim neck; (**b**) with a slim neck.

**Figure 9 entropy-26-00591-f009:**
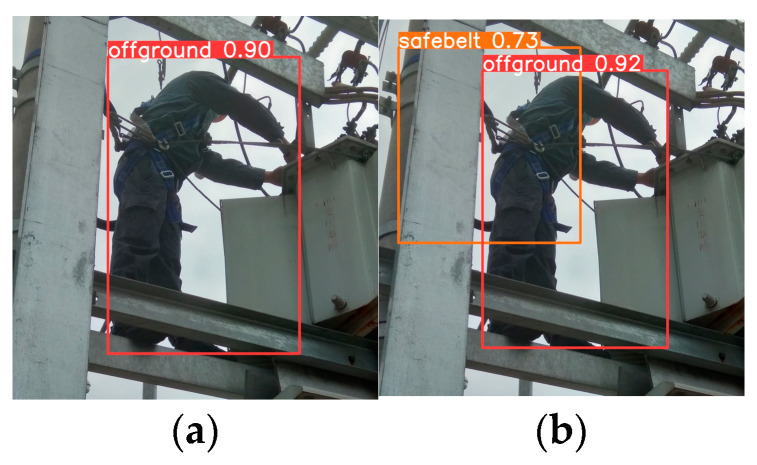
Impact of loss functions on detection performance. (**a**) CIoU; (**b**) EIoU.

**Table 1 entropy-26-00591-t001:** Comparison of model performance.

Model	mAP50	mAP50-95	Parameters	FLOPs
YOLOv5	91.2%	66.3%	7.2 M	15.8 G
YOLOv8	91.4%	74.8%	11.1 M	28.4 G
YOLOv6 3.0	89.1%	65%	18.5 M	45.3 G
Gold-YOLO	91.8%	73.3%	21.5 M	46.0 G
YOLOX	92.9%	67.7%	9.0 M	26.8 G
ATSS	76.7%	43%	31.9 M	201.6 G
ESE-YOLOv8 (Ours)	92.7%	75.7%	10.3 M	26.1 G

**Table 2 entropy-26-00591-t002:** Ablation study of ESE-YOLOv8.

EMA	EIoU	Slim-Neck	mAP50	mAP 50–95	Parameters	FLOPs
√	√	√	92.7%	75.7%	10.3 M	26.1 G
√	√	×	92.4%	75.7%	11.2 M	29.4 G
√	×	×	92.1%	75.5%	11.2 M	29.4 G
×	×	×	91.4%	74.8%	11.1 M	28.4 G

## Data Availability

The data presented in this study are available on request from the corresponding author due to privacy and confidentiality concerns, as the dataset used in this study contains a substantial number of identifiable human images.

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
