# Peer review of "ESE-YOLOv8: A Novel Object Detection Algorithm for Safety Belt Detection during Working at Heights"

_entropy, 2024, doi:10.3390/e26070591_

Round 1

Reviewer 1 Report

Comments and Suggestions for Authors

This manuscript describes an architecture ESE-YOLOv8 for safety belt detection. The following comments are advised to be addressed:

- Section 3.2, Line 173, "mitigate the adverse effects of dimensionality reduction...", please explain more on the "adverse effects".

- Section 3.2, Line 176-191. This description should be revised to focus only on key features of EMA and how those key features can "mitigate the adverse effects of dimensionality reduction..."

- Line 219-221, please elaborate more on how the computational load is enhanced. How much gain of computational efficiency can be expected?

- Section 4.2, Eq. (10). is R differentiable?

- Title of Section 3.2, it would be better to use the full form and abbreviation: Efficient Multi-Scale Attention (EMA). Figure 2 shows that the resolution is too low.

Author Response

Comments 1: Section 3.2, Line 173, "mitigate the adverse effects of dimensionality reduction...", please explain more on the "adverse effects".

Response 1: We apologize for not providing detailed explanations. We have added relevant explanations along with a reference to help clarify this point. The new content can be found in Section 3.2, lines 202-209.

Comments 2: Section 3.2, Line 176-191. This description should be revised to focus only on key features of EMA and how those key features can "mitigate the adverse effects of dimensionality reduction..."

Response 2: We apologize for the lack of clarity and have revised the relevant section. Please allow us to keep the explanation of the internal structure of EMA, as it may be helpful for understanding its operation. Regarding how it "mitigates the adverse effects of dimensionality reduction," EMA achieves cross-channel interaction through its unique structure and avoids dimensionality reduction during the attention mechanism, thus preventing the associated negative effects. The new content can be found in Section 3.2, lines 221-224.

Comments 3: Line 219-221, please elaborate more on how the computational load is enhanced. How much gain of computational efficiency can be expected?

Response 3: We apologize for the incomplete information. We have added some explanations. According to the reference, GSConv reduces the computational load by 50% compared to standard convolution operations. The new content can be found in Section 3.2, lines 251-260.

Comments 4: Section 4.2, Eq. (10). is R differentiable?

Response 4: We apologize for the misleading equation. R is not differentiable because it is derived from a curve made up of discrete points. Our intention was to express that AP is the area under the P-R curve, so we used an integral expression to convey the geometric meaning of AP. We have now changed it to a discrete expression. The new content can be found in Section 4.2, equation (10).

Comments 5: Title of Section 3.2, it would be better to use the full form and abbreviation: Efficient Multi-Scale Attention (EMA). Figure 2 shows that the resolution is too low.

Response 5: We have revised the title of Section 3.2 to include the full form and abbreviation: Efficient Multi-Scale Attention (EMA). We have also replaced Figure 2 with a higher resolution image. Thank you for your feedback. The new content can be found in Section 3.2, lines 200 and lines 223.

Reviewer 2 Report

Comments and Suggestions for Authors

This article describes an entire framework for the supervision of workers who are performing their activity correctly, or not, in the work at height environment. The framework is based on deep learning, integrating different techniques that, in sum, improve the results obtained by the popular YOLO proposal. The whole article is well organised and well written, making it easy to read. A brief description of the contributions, even if they are related to previous proposals (EMA, Slim-neck, CIoU/EIoU), would be appreciated in the introduction.

As a minor comment, Figure 1 would need to be modified to make the text readable.

Author Response

Comments 1: This article describes an entire framework for the supervision of workers who are performing their activity correctly, or not, in the work at height environment. The framework is based on deep learning, integrating different techniques that, in sum, improve the results obtained by the popular YOLO proposal. The whole article is well organised and well written, making it easy to read. A brief description of the contributions, even if they are related to previous proposals (EMA, Slim-neck, CIoU/EIoU), would be appreciated in the introduction.

Response 1: Thank you for your feedback. We have revised the introduction to make the contributions clearer and more readable. The relevant content can be found in Section 1, lines 67-91.

Comments 2: As a minor comment, Figure 1 would need to be modified to make the text readable.

Response 2: We apologize for the poor quality of the original image. We have replaced it with a new image that has a more compact structure and larger font size to improve readability. The updated content can be found in Section 3, line 156.